# Ovarian Cancer Stemness: Biological and Clinical Implications for Metastasis and Chemotherapy Resistance

**DOI:** 10.3390/cancers11070907

**Published:** 2019-06-28

**Authors:** Takeshi Motohara, Hidetaka Katabuchi

**Affiliations:** Department of Obstetrics and Gynecology, Faculty of Life Sciences, Kumamoto University, Honjo 1-1-1, Kumamoto City, Kumamoto 860-8556, Japan

**Keywords:** ovarian cancer, cancer stem cell, metastasis, chemotherapy resistance

## Abstract

Epithelial ovarian cancer is a highly lethal gynecological malignancy that is characterized by the early development of disseminated metastasis. Though ovarian cancer has been generally considered to preferentially metastasize via direct transcoelomic dissemination instead of the hematogenous route, emerging evidence has indicated that the hematogenous spread of cancer cells plays a larger role in ovarian cancer metastasis than previously thought. Considering the distinctive biology of ovarian cancer, an in-depth understanding of the biological and molecular mechanisms that drive metastasis is critical for developing effective therapeutic strategies against this fatal disease. The recent “cancer stem cell theory” postulates that cancer stem cells are principally responsible for tumor initiation, metastasis, and chemotherapy resistance. Even though the hallmarks of ovarian cancer stem cells have not yet been completely elucidated, metastasized ovarian cancer cells, which have a high degree of chemoresistance, seem to manifest cancer stem cell properties and play a key role during relapse at metastatic sites. Herein, we review our current understanding of the cell-biological mechanisms that regulate ovarian cancer metastasis and chemotherapy resistance, with a pivotal focus on ovarian cancer stem cells, and discuss the potential clinical implications of evolving cancer stem cell research and resultant novel therapeutic approaches.

## 1. Introduction

Epithelial ovarian cancer is the most lethal cause of death among gynecological malignancies and is characterized by an early metastatic spread throughout the peritoneal cavity, along with extensive disseminated tumors, omental caking, and accumulation of malignant ascites [1,2]. The high mortality and low survival rate in ovarian cancer can be attributed to its nonspecific symptoms that generally manifest only after progression to an advanced stage, as well as the fact that only few women are diagnosed before cancer metastasis to the peritoneal cavity or distant parenchymal organs [3,4,5]. Though most patients with the advanced stage of the disease may respond to a combination of taxane and platinum-based chemotherapy, chemoresistant residual cancer cells can persist in metastatic sites, where they remain dormant for prolonged periods after initial therapy and eventually lead to a relapse [6,7,8]. Despite ongoing efforts in developing extirpative surgery and intensive combination chemotherapy, long-term clinical outcomes in patients with advanced ovarian cancer have not significantly improved over the recent decades [9,10,11].

The cancer stem cell theory has been recently postulated and states that cancers are composed of hierarchies of cells sustained by tumor-initiating cells, conceptually termed cancer stem cells, with distinct phenotypes and a high tumorigenic potential [12,13]. Inherently, cancer stem cells possess the ability to self-renew and differentiate into multiple lineages of differentiated cancer cells, thereby establishing both a phenotypic and functional heterogeneity in the hierarchical organization of tumors [14,15]. Additionally, a large body of research indicates that a subpopulation of cancer stem cells is practically responsible for driving metastasis, therapy resistance, and the relapse of cancers [16,17]. In regard to ovarian cancer, Bapat et al. provided the first evidence for the existence of ovarian cancer stem cells in the malignant ascites in 2005; thereafter, extensive efforts have been made to evaluate the biological mechanisms that regulate ovarian cancer stemness [18]. Clinically, ovarian cancer stem cells have been shown to survive conventional chemotherapy, which generally targets rapidly dividing cancer progenitor cells or more differentiated cancer cells. This small population of cancer stem cells then gives rise to chemoresistant recurrent tumors at metastatic sites [9,19]. Given the distinct biology of ovarian cancer stem cells, an elucidation of the cellular and molecular mechanisms underlying cancer metastasis and chemotherapy resistance, with a central focus on the ovarian cancer stem cells, will shed light on novel therapeutic strategies that can improve clinical outcomes in patients with advanced ovarian cancer.

Here, we provide an overview of the crucial role of ovarian cancer stem cells in tumor initiation, metastasis, and resistance to chemotherapy, and we also highlight the biological mechanisms that regulate ovarian cancer metastasis and chemoresistance, with a particular focus on the various stem cell signaling pathways. Furthermore, the potential clinical implications of novel therapeutic avenues that target cancer stem cells in patients with ovarian cancer are discussed. We hope that this review will provide a comprehensive understanding of the molecular basis of ovarian cancer stem cells and will guide clinical decision-making, the development of novel therapeutic interventions, and the better management of patients with ovarian cancer.

## 2. Identification and Characterization of the Ovarian Cancer Stem Cell Population

The clinical course of ovarian cancer treatment typically results in a vast majority of patients undergoing relapse with chemoresistant metastatic tumors despite high response rates to first-line chemotherapy. In this regard, even though ovarian cancer stem cells have not been completely elucidated [20,21,22], a small population of metastasized cancer cells with a high degree of chemoresistance may possess cancer stem cell properties and play a crucial role in relapse at metastatic sites [23,24,25]. Therefore, elucidating the biology of ovarian cancer stem cells will yield novel insights into the still elusive mechanisms of metastasis and chemotherapy resistance in ovarian cancer.

In recent years, various specific markers of stemness, including CD44, CD117, aldehyde dehydrogenase (ALDH), CD133, CD24, and the epithelial cell adhesion molecule (EpCAM), have been used to isolate and characterize ovarian cancer stem cells [26,27,28,29,30,31,32] (Table 1). Zhang et al. identified a self-renewing subpopulation of ovarian cancer cells that can serially propagate their original tumor phenotype in an in vivo mouse model [33]. Additionally, the authors found that sphere-forming cells co-express CD44 and CD117 and that those two specific markers can be used to isolate highly tumorigenic ovarian cancer stem cells [33]. Similarly, Silva et al. demonstrated that ALDH and CD133 expression can be used to define distinct heterogeneous subpopulations of ovarian cancer stem cells. By using an in vivo mouse model, they showed that ALDH enzymatic activity and CD133 positivity were closely correlated with ovarian cancer stem cell phenotypes with high tumorigenic potential [34]. In addition, Gao et al. isolated a series of cancer cell clones from ovarian cancer specimens and identified a subpopulation enriched for ovarian cancer stem cells defined by CD24 expression. The authors also found that CD24-positive ovarian cancer cells preferentially express higher levels of stem cell genes—including Nestin, β-catenin, Bmi-1, Oct4, Oct3/4, Notch1, and Notch4—than the CD24-negative counterpart, and they contribute to a specific capacity for self-renewal and multi-differentiation [35]. In a related move, our research group previously generated mouse ovarian tumor-initiating cells by the siRNA-mediated knockdown of tumor suppressor p53 followed by the transduction of c-Myc and K-Ras oncogenes [36]. Intriguingly, we also identified a subpopulation of EpCAM-positive cancer cells as possible candidates for ovarian cancer stem cells in an established mouse ovarian cancer model. By using an in vivo limiting dilution assay, we found that EpCAM-positive cancer cells isolated from hierarchically organized ovarian tumors have greater tumor-initiating properties than EpCAM-negative cancer cells. Additionally, with respect to the differentiation capacity of EpCAM-positive cancer cells, we showed that these cells can give rise to less tumorigenic EpCAM-negative cancer cells, attesting to the multi-lineage differentiation potential of these cells [36].

Taken together, it appears that there are currently a significant number of markers that can be used to isolate a specific ovarian cancer stem cell population; however, these markers have not proven to be ubiquitously expressed in a given tumor, and it remains challenging to identify bona fide ovarian cancer stem cells. In view of this, further studies are needed to consistently enrich the subpopulation of ovarian cancer stem cells and to precisely explain the biology of ovarian cancer stem cells (Table 1).

## 3. High Metastatic Potential of Ovarian Cancer Stem Cells

Ovarian cancer has been generally considered to preferentially metastasize via transcoelomic rather than hematogenous dissemination [55,56,57]. Transcoelomic metastasis is implicated in the exfoliation of cancer cells from primary ovarian tumors, their survival as multicellular spheroids that float in the ascitic fluid, and their subsequent metastatic colonization in the peritoneal cavity [55,58,59,60] (Figure 1). Previous studies demonstrated that ovarian cancer stem cells play a pivotal role in the formation of multicellular spheroids during the transcoelomic peritoneal dissemination of ovarian cancer [61,62]. Liao et al. showed that spheroids in the malignant ascites are enriched in cancer stem cells that are primarily responsible for ovarian cancer tumorigenesis, progression, and transcoelomic metastasis [41]. The authors also revealed that these sphere-forming cancer cells express stemness-related genes—such as Notch1, Nanog, Cdcp1, and Myc—and upregulate CD117 expression and ALDH activity, thereby promoting the development of peritoneal and omental metastasis in an in vivo ovarian cancer xenograft model [41]. More recently, Roy et al. elucidated the functional role of CD133 expression in the early steps of peritoneal metastasis. Using an ex vivo peritoneal explant adhesion assay, they demonstrated that CD133 expression significantly increased the adhesion and invasion of ovarian cancer cells to the peritoneal mesothelium, suggesting that CD133-positive ovarian cancer cells are involved in communicating with the microenvironmental niche in the peritoneum [46]. In this regard, a previous study identified a key role for peritoneal mesothelial cells in the maintenance of ovarian cancer stemness, thus raising the possibility that the peritoneal microenvironment has the potential to form a cancer stem cell niche for ovarian cancer dissemination [63]. In addition, a recent study by Burgos-Ojeda et al. showed that CD24-positive ovarian cancer cells manifest increased levels of STAT3 phosphorylation compared with CD24-negative cells, an increase which is associated with enhancing disseminated metastasis [47]. More importantly, the blockade of STAT3 phosphorylation with a JAK 2 inhibitor significantly reduced ovarian cancer metastasis in an established genetic mouse model of ovarian cancer [47]. On another front, our study group demonstrated a crucial role for CD44 variant 6 (CD44v6) in the regulation of various biological functions during the transcoelomic metastasis of ovarian cancer [37]. A clinicopathological analysis of CD44v6 expression indicated that peritoneal disseminated tumors are highly enriched in CD44v6-positive ovarian cancer cells compared with corresponding primary tumors, suggesting that CD44v6-positive cancer cells are associated with peritoneal transcoelomic dissemination and that the peritoneum may function as a metastatic microenvironmental niche that contributes to the process of metastatic colonization in the peritoneal cavity. Intriguingly, and consistent with these clinical observations, a subpopulation of CD44v6-positive cancer cells gave rise to extensive disseminated metastatic tumors in the peritoneal cavity, whereas CD44v6-negative cancer cells manifested a lower metastatic ability in an in vivo mouse model. Furthermore, a limiting dilution assay demonstrated that CD44v6-positive cancer cells showed a greater tumor initiating capability than CD44v6-negatice cancer cells, indicating that the subpopulation of CD44v6-positive cancer cells may serve as specialized metastasis-initiating cells within the intraperitoneal milieu [37] (Figure 1 and Table 1).

Generally, although transcoelomic dissemination has been thought to be the major route of ovarian cancer metastasis [60], a recent study by Pradeep et al. revealed novel mechanisms of hematogenous metastasis to the omentum followed by intraperitoneal disseminated metastasis in a parabiosis mouse model that allowed for the sharing of blood circulation [64]. They showed that circulating ovarian cancer cells derived from the host mouse can first metastasize to the omentum of the conjoined guest mouse via a hematogenous route and subsequently spread to the peritoneal cavity, thus shifting the paradigm in mechanisms that regulate the disseminated metastasis of ovarian cancer in the peritoneal cavity. Mechanistically, the ErbB3/Neuregulin-1 signaling axis has been shown to play a functional role in such hematogenous ovarian cancer metastasis with a strong tropism toward the omentum [64]. Additionally, with respect to distant parenchymal metastasis via the hematogenous route in ovarian cancer, our recent study demonstrated that CD44v6-positive ovarian cancer cells represent a central player in the development of distant metastasis in parenchymal organs [38] (Figure 1). Clinicopathological evidence corroborated the fact that ovarian cancers with greater numbers of CD44v6-positive cancer cells were associated with higher rates of distant metastasis at the time of ovarian cancer diagnosis. Furthermore, an immunohistochemical analysis detected a significantly higher percentage of CD44v6-positive cells in distant metastatic tumors than primary ovarian tumors, suggesting that CD44v6-positive cancer cells play a key role in driving distant metastasis via a hematogenous spread. More importantly, a Kaplan–Meier analysis showed that distant metastasis-free survival was significantly different between CD44v6-high and -low groups, indicating that CD44v6 expression is involved in greater distant metastatic relapse in patients with stage I–III ovarian cancer. It should be noted here that a multivariate analysis identified CD44v6 expression as an independent risk factor for distant metastatic relapse, implying that CD44v6 expression may be a crucial predictive biomarker for distant parenchymal metastasis in ovarian cancer patients [38]. A number of studies of late years have indicated that ALDH activity is closely linked to both ovarian cancer stemness and metastasis [65,66]. Bai et al. showed that epidermal growth factor-like domain 6 (EGFL6), which acts as a stem cell regulatory factor, promotes the asymmetric division of ALDH-positive ovarian cancer stem cells and thereby increases cancer cell proliferation in vitro and tumor growth in vivo [43]. Interestingly, the authors also found that vascular EGFL6 expression enhances the distant metastasis of ovarian cancer cells through the hematogenous route, whereas an EGFL6 blockade reduces the hematogenous spread of ovarian cancer cells to distant parenchymal organs [43]. These findings provide a therapeutic rationale for targeting specific molecules in the microenvironmental niche (Figure 1 and Table 1).

## 4. Resistance to Chemotherapy in Metastasized Ovarian Cancer Stem Cells

The standard treatment for patients with ovarian cancer is a maximum debulking surgery followed by the administration of taxane and platinum-based chemotherapy [6,67]. Growing evidence indicates that the administration of chemotherapy can significantly and efficiently shrink tumor mass, while these chemotherapeutic agents induce the enrichment of cancer stem cells that are associated with a future tumor relapse at metastatic sites [20,68] (Figure 2). Thus, uncovering the molecular events underlying chemotherapy resistance with respect to the survival of metastasized ovarian cancer stem cells can significantly improve clinical outcomes in patients with advanced ovarian cancer [69].

A previous study revealed that treatment with cisplatin results in the enhanced expression of cancer stem cell markers—including CD44, α2 integrin subunit, CD117, CD133, and EpCAM—and the upregulation of stem cell factors—such as Nanog and Oct4—in residual ovarian cancer cells in in vitro assays [70]. In addition, a recent study by Gao et al. demonstrated that the immunohistochemical expression of CD44 is significantly higher in both metastatic and recurrent ovarian tumor tissues than patient-matched primary tumor tissues; this study also demonstrated that the increased expression of CD44 is involved in poor survival outcomes in ovarian cancer patients. Intriguingly, using an ovarian cancer xenograft model, they showed that paclitaxel treatment promotes the enhanced expression of CD44 in recurrent tumors, indicating that a subpopulation of CD44-positive cancer cells possesses the ability to increase chemotherapy resistance at metastatic sites [39]. In a related study, Steg et al. reported that metastatic ovarian tumors obtained from recurrent platinum-resistant patients are more densely composed of the subpopulation of cancer stem cells that express CD44, ALDH1A1, and CD133, in comparison with primary ovarian tumors [40]. These findings support the theoretical view that metastasized ovarian cancer stem cells are implicated in composing chemoresistant populations that exist at the end of initial therapy for patients with ovarian cancer. Furthermore, the authors revealed functional roles for several stem cell-related signaling pathways, such as TGF-β co-receptor endoglin and the hedgehog transcriptional mediator Gli2, in regulating chemotherapy resistance in ovarian cancer stem cells [40], thus raising the possibility that novel therapeutic strategies targeting these pathways in metastasized ovarian cancer stem cells would be attractive approaches to overcome chemotherapy resistance. More recently, we also demonstrated a crucial role for EpCAM in regulating platinum-based chemotherapy resistance in patients with ovarian cancer [48]. A clinicopathological analysis showed that EpCAM expression is significantly higher in tumor tissues from patients who received platinum-based chemotherapy than in corresponding tumor tissues before chemotherapy. Furthermore, evidence from a multivariate analysis points to the fact that the immunohistochemical expression of EpCAM is an independent risk factor for the development of resistance to platinum-based chemotherapy, indicating that EpCAM expression may be a potential predictive biomarker of chemotherapeutic response in ovarian cancer patients. Consistent with clinical data, we also found that a subpopulation of EpCAM-positive cancer cells manifests significantly a higher viability after cisplatin treatment by preventing chemotherapy induced-apoptosis, which is regulated by the EpCAM-Bcl-2 signaling axis, in in vitro assays. It is intriguing to note that in an in vivo mouse ovarian cancer model, a treatment with platinum agents caused significant tumor shrinkage; in contrast, a substantial enrichment of EpCAM-positive ovarian cancer cells was also observed, demonstrating that the remaining subpopulation of EpCAM-positive cancer stem cells is intimately associated with tumor relapse after chemotherapy [48] (Figure 2).

In recent years, a number of studies have focused on assessing the potential role of neoadjuvant chemotherapy (NAC) in patients with advanced ovarian cancer; however, the precise effectiveness of NAC remains highly controversial in clinical practice [71,72,73]. With regard to the relationship between NAC and chemoresistant ovarian cancer stem cells at metastatic sites, multiple studies have shown that NAC is correlated with the enrichment of ovarian cancer stem cells that contributes to microscopic residual disease at metastatic sites, eventually causing a relapse [73]. Ayub et al. showed that ovarian cancer stem cells, defined by their high ALDH1 activity, preferentially survive and increase in number after NAC and that such enrichment of ALDH1 expression is associated with unfavorable survival outcomes in patients with advanced ovarian cancer [44]. Taken together, this evidence supports the premise that NAC is a major driver of chemotherapy resistance mediated by ovarian cancer stem cells [74]. With regard to the indications for NAC in patients with advanced ovarian cancer, gynecologists need to determine an appropriate therapeutic strategy on the basis of the fact that NAC is closely correlated with inducing the enrichment of ovarian cancer stem cells at metastatic sites. Further basic and clinical investigations are required to clarify the survival impact of NAC on cancer stem cell biology in the management of patients with advanced ovarian cancer (Table 1).

## 5. Embryonic Developmental Signaling Pathways in Chemoresistant Ovarian Cancer Stem Cells

Conceptually, it is possible that targeting the molecular pathways involved in the metastasis of chemoresistant ovarian cancer stem cells will provide novel treatment modalities for ovarian cancer [9,75]. A large number of studies have demonstrated that cancer stem cells manifest fundamental biological similarities with normal stem cells at the cellular and molecular level [12]. In fact, highly conserved signal transduction pathways implicated in embryonic development, tissue homeostasis, and stem cell biology, such as Wnt, Notch, and Hedgehog, contribute to the regulation of the self-renewal and differentiation of cancer stem cells in several types of malignancies [76] (Figure 2).

The canonical Wnt signaling pathway mediated through β-catenin is highly regulated for maintenance of biological homeostasis, and this pathway is aberrantly activated in numerous solid cancers [77,78]. Importantly, gene expression studies from The Cancer Genome Atlas (TCGA) have identified altered Wnt/β-catenin components in ovarian cancer with an unfavorable prognosis, suggesting that the Wnt/β-catenin pathway plays a functional role in regulating ovarian cancer biology [49]. With respect to the correlation between Wnt/β-catenin signaling and ovarian cancer stem cells, Cau et al. provided evidence for the functional roles of stem cell factor receptor CD117 in the tumorigenic properties and resistance to chemotherapeutic drugs through the activation of the Wnt/β-catenin signaling pathway [42]. The authors also showed that the siRNA-mediated knockdown of CD117 expression markedly reduced the number and size of ovarian cancer stem cell subpopulations and the pro-tumorigenic activity of ovarian cancer cells. In terms of molecular mechanisms underlying chemotherapy resistance, they revealed that CD117 overexpression induces the upregulation of ATP-binding cassette G2 (ABCG2) through the Wnt/β-catenin signaling pathway, thereby enhancing the chemotherapy resistance of ovarian cancer cells in a hypoxic tumor microenvironment [42]. More recently, Nagaraj et al. demonstrated that the Wnt/β-catenin signaling pathway contributes to maintain cancer stem cell characteristics and platinum resistance in ovarian cancer cells. By using patient-derived xenograft (PDX) models of ovarian cancer, the authors revealed that the expression of various Wnt/β-catenin target genes, including Axin2, DKK2, Lef1, and Lgr5, and the expression of the Wnt ligand WNT5A are upregulated in platinum-resistant PDX-derived cancer cells, compared with platinum-sensitive cancer cells [45]. Furthermore, platinum-resistant ovarian cancer cells showed a higher expression of several cancer stem cell markers, such as ALDH1, CD24, and EpCAM, which are known direct targets of Wnt/β-catenin signaling. More importantly, with regard to Wnt-targeted therapeutic strategies, the Wnt/β-catenin specific inhibitor, iCG-001, significantly reduced the frequency of cancer stem cell subpopulations and sensitized ovarian cancer cells to cisplatin, suggesting that the Wnt/β-catenin signaling blockade can suppress ovarian cancer stem cell characteristics and induce chemosensitivity in ovarian cancer cells [45]. Taken together, it appears that targeting stem cell pathways would be an effective treatment option for metastasized ovarian cancer (Figure 2 and Table 1).

Notch is an evolutionarily conserved signaling pathway that links complex processes such as embryonic development and stem cell maintenance [79,80]. In the Notch signaling cascade, the activation of Notch receptors by their ligands leads to the proteolytic cleavage of the Notch intracellular domain, mediated by γ-secretase, and the Notch intracellular domain translocates into the nucleus to regulate gene transcription [80,81]. Recently, increasing evidence has indicated that the Notch signaling pathway may have a key role in cancer stem cell function in a variety of cancers [82,83]. A previous study reported that Notch3 protein levels are elevated in recurrent tumor tissues, compared with primary tumor tissues from same patients, and that increased Notch3 expression is significantly correlated with poor clinical outcomes in ovarian cancer patients [50]. Interestingly, the expression of the Notch3 intracellular domain can enhance the expression of various embryonic stem cell markers, including Oct4, Nanog, Klf4, Rex1, Rif1, Sall4, and NAC1, as well as the ATP-binding cassette sub-family B member 1 (ABCB1), and can thereby regulate the development of resistance to carboplatin in ovarian cancer cells [50]. Additionally, Shannon et al. demonstrated that Notch signaling plays a crucial role in the maintenance of ovarian cancer stem cells and resistance to platinum therapy in in vitro assays and in vivo mouse models [51]. The authors also showed that Notch3 overexpression in ovarian cancer cells results in the expansion of a subpopulation of cancer stem cells, defined as a side-population of cells, and that it contributes to a significant increase in resistance to cisplatin. More importantly, they found that a Notch pathway inhibitor, γ-secretase inhibitor, preferentially attenuates cancer stem cell properties in ovarian cancer cells, and that a combination therapy of γ-secretase inhibitor and cisplatin has a synergistic cytotoxic effect on Notch-dependent cancer cells by inducing a DNA-damage response, a G_2_/M cell-cycle arrest, and an increased apoptotic cell death. These findings suggest that combination therapy that targets both cancer stem cells and the bulk of cancer cells would be critical for eliminating ovarian cancer [51,84] (Figure 2 and Table 1).

The hedgehog signaling transduction pathway is also a crucial regulator of stem cell fate during embryonic development, and its deregulation is involved in cancer development, proliferation, survival, and metastasis [85,86]. The components of the hedgehog pathway include three secreted ligands (Sonic Hedgehog, Indian Hedgehog, and Desert Hedgehog), the hedgehog receptors Patched (PTCH1 and PTCH2) and smoothened (SMO), and the Gli transcription factors (Gli1, Gli2, and Gli3) [87,88,89]. Growing evidence has indicated that the activation of the hedgehog signaling pathway is intimately associated with the regulation of cancer stem cell phenotypes, including self-renewal, differentiation, and tumor initiation [76,90]. With regard to the relationship between the hedgehog pathway and ovarian cancer stem cells, a previous in vitro study revealed that ovarian cancer cells exhibit an increased intracellular Gli1 expression that is correlated with the increased formation of multicellular spheroids with cancer stem cell properites [52]. Furthermore, Chen et al. demonstrated that Gli1 plays an important role in the regulation of ABC transporters ABCB1 and ABCG2 by directly binding to their promoter regions, and it thereby enhances resistance to cisplatin and paclitaxel in spheroid-forming ovarian cancer cells [53]. More recently, Song et al. investigated the correlation between the expression of the hedgehog signaling pathway components and clinicopathological features in different types of ovarian epithelial tumor tissues, including benign, borderline, and malignant tumors [54]. This study identified a stronger expression of SMO and Gli1 in borderline and malignant tumor tissues compared with normal ovarian epithelial or benign tumor tissues. Furthermore, the authors showed that the expression levels of SMO and Gli1 in cancer tissues from patients with cisplatin-resistant ovarian cancer were significantly higher than those from patients with cisplatin-sensitive ovarian cancer, suggesting that the activation of hedgehog signaling is important for malignant transformation and chemotherapy resistance [54]. Collectively, these findings indicate that hedgehog signaling also has therapeutic potential with respect to overcoming resistance to chemotherapy in ovarian cancer [88,91] (Figure 2 and Table 1).

## 6. Conclusions and Future Perspectives

Over the last couple of decades, cancer stem cell research has evolved remarkably, and it has integrated knowledge from a wealth of basic research and clinical studies on cancer stem cell biology, all of which have provided cause for optimism for the development of more effective cancer therapies [92,93]. Nevertheless, most of these significant findings have not yet been translated into meaningful clinical endpoints in various types of malignancies [93,94]. Therefore, it is truly expected that the translation of cancer stem cell research into diagnostics and therapeutics will lead to an improvement in the management of cancer patients [94,95,96].

Considering the fact that the majority of ovarian cancer-related deaths are caused by the metastatic dissemination of cancer cells, an in-depth understanding of ovarian cancer metastasis is crucial to overcome ovarian cancer [38]. As for the current therapeutic strategies in the management of metastatic ovarian cancer, the treatment efficacy of conventional cytotoxic chemotherapies and molecular targeted therapies is severely limited, because metastasized ovarian cancer stem cells manifest enhanced chemoresistance by regulating various signaling pathways implicated in stem cell biology [97,98]. In view of this, the identification and validation of multiple therapeutic approaches, including cancer stem cell markers and stem cell signaling pathways, should help in the development of more effective therapeutic modalities for improving clinical outcomes in patients with ovarian cancer [92,99,100,101].

In conclusion, toward the ultimate goal of higher survival rates and brighter prognoses for patients with advanced ovarian cancer, further research on the evolving molecular biology of ovarian cancer stem cells in association with cancer metastasis and resistance to chemotherapy will be required to not only broaden our current understanding of the hallmarks of ovarian cancer but to also shed light on ways to develop innovative strategies for the diagnosis and treatment of this life-threatening malignancy.

## Figures and Tables

**Figure 1 cancers-11-00907-f001:**
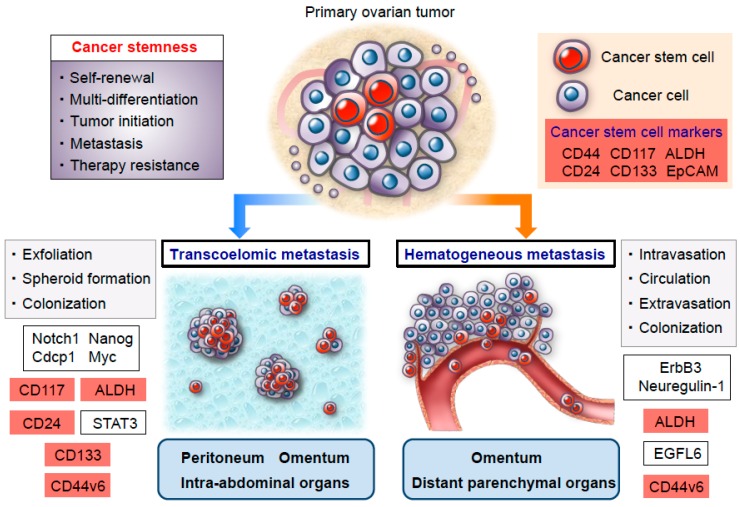
Schema of an ovarian cancer stem cell model for transcoelomic and hematogenous metastasis. A subpopulation of ovarian cancer stem cells has greater tumor-initiating properties and serves as functionally and molecularly distinct metastasis-initiating cells. Especially, ovarian cancer stem cells are responsible for not only driving transcoelomic dissemination but also hematogenous metastasis via the activation of various signaling pathways involved in stem cell biology. Cancer stem cell markers are highlighted in red.

**Figure 2 cancers-11-00907-f002:**
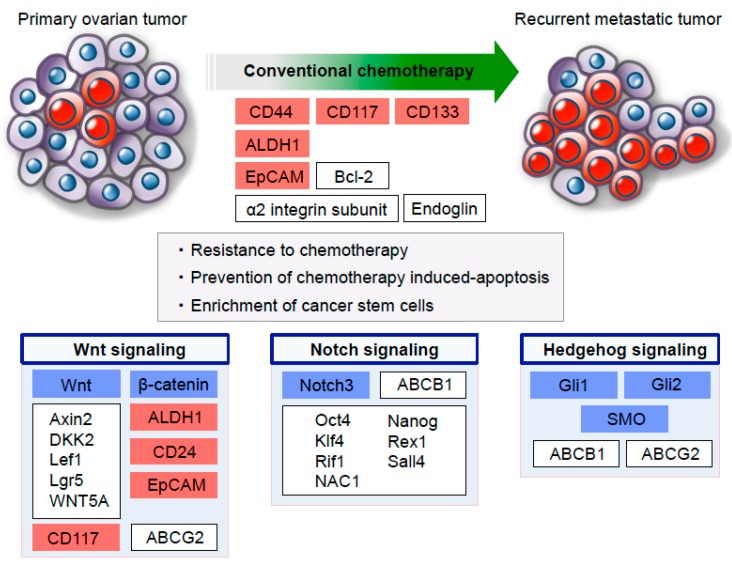
Schema of an ovarian cancer stem cell model for chemotherapy resistance. A subpopulation of ovarian cancer stem cells is able to survive conventional chemotherapy and is closely correlated with the development of recurrent metastatic tumors after chemotherapy. In fact, chemotherapeutic drugs induce the enrichment of ovarian cancer stem cells at metastatic sites, and metastasized ovarian cancer stem cells manifest enhanced chemoresistance by regulating various stem cell signaling pathways, including Wnt, Notch, and Hedgehog. Cancer stem cell markers are highlighted in red, and embryonic developmental signaling pathways are highlighted in blue.

**Table 1 cancers-11-00907-t001:** Ovarian cancer stem cell markers and stem cell signaling pathways.

Marker/Signaling Pathway	Type of Protein	Function in Ovarian Cancer Stem Cell	Reference
CD44 (CD44v6)	Glycosylated transmembrane receptor	Tumor initiation, transcoelomic metastasis, hematogenous metastasis, chemoresistance	[27,30,33,37,38,39,40]
CD117	Tyrosine kinase receptor	Tumor initiation, transcoelomic metastasis, chemoresistance	[33,41,42]
ALDH	Enzyme responsible for oxidizing intracellular aldehydes	Tumor initiation, transcoelomic metastasis, hematogenous metastasis, chemoresistance	[29,31,34,40,41,43,44,45]
CD133	Pentaspan transmembrane glycoprotein	Tumor initiation, adhesion, invasion, transcoelomic metastasis, chemoresistance	[28,29,32,34,40,46]
CD24	Mucin-like cell surface glycoprotein	Tumor initiation, self-renewal, multi-differentiation, transcoelomic metastasis, chemoresistance	[27,35,45,47]
EpCAM	Type I transmembrane glycoprotein	Tumor initiation, multi-differentiation, sphere formation, chemoresistance, prevention of chemotherapy induced-apoptosis	[27,36,45,48]
Wnt	Palmitoylated secreted glycoprotein	chemoresistance	[42,45,49]
Nothch	Type I transmembrane glycoprotein	chemoresistance	[50,51]
Hedgehog	Secreted signaling protein	chemoresistance	[52,53,54]

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
