# Peer review of "Ovarian Cancer Stemness: Biological and Clinical Implications for Metastasis and Chemotherapy Resistance"

_cancers, 2019, doi:10.3390/cancers11070907_

Reviewer 1 Report

This is a well-written manuscript which focuses on the current knowledge on the role of cell stemness in ovarian cancer. The authors thoroughly reviewed proved and/or potential contribution of ovarian cancer stems cells (OCSCs) to differential steps of ovarian cancer progression, metastasis and chemoresistance acquisition.  At the end, the researchers discuss several developmental signaling pathways which look important for chemoresistant OCSCs and could potentially be used for therapeutic implications. Well-compiled figures are a plus.

I have several suggestions and minor comments to the authors:

1.       It would be helpful to generate a table at the end of the article which summarizes established/putative OCSC stem markers and stem factors and their implications in particular steps of ovarian cancer metastasis and chemoresistance with relevant citations in the last column (all that were mentioned throughout the manuscript text body).

2.       Minor grammatical error and typos throughout text should be eliminated (lines 93, 97, 98, 123, 138, 201, 237-238, 240-241, 246, 322).

3.       Some of the abbreviations used in the text are not introduced at first mention, please correct.

4.       Line 150: the referral to “Figure 1” in this context makes an impression that this figure demonstrates the study by Pradeep et al, which is not the case. It would be better to introduce Figure 1 of the current manuscript in a more suitable place in order not to mislead the readers.

5.       Line 226: “Ayub et al. have shown that ovarian cancer stem cells ….”  - lacks a citation. Neither reference 64 no 65 of the bibliography has Ayub as the first author.

6.       It is better to re-introduce same citations at the end of each sentence where a certain study is mentioned. Authors often describe concrete studies in multiple sentences and do not cite them until the end of the paragraph. To a reader, it is not clear whether the statements are authors’ own thoughts vs the findings belong to previously or subsequently cited paragraph vs the citations are simply missing.

Author Response

Reviewer #1 (Reviewer Comments to the Author):

1.     It would be helpful to generate a table at the end of the article which summarizes established/putative OCSC stem markers and stem factors and their implications in particular steps of ovarian cancer metastasis and chemoresistance with relevant citations in the last column (all that were mentioned throughout the manuscript text body).

As per the reviewer’s recommendation, I would totally agree with your idea. However, I know similar table in previous review article reported by Roy et al. (Cancers 2018, 10, 0241; doi:10.3390/cancers10080241). Therefore, we decided not to add new table. Thank you very much for your constructive comments and suggestions.

2.     Minor grammatical error and typos throughout text should be eliminated (lines 93, 97, 98,

123, 138, 201, 237-238, 240-241, 246, 322).

3.     Some of the abbreviations used in the text are not introduced at first mention, please

correct.

4.     Line 150: the referral to “Figure 1” in this context makes an impression that this figure

demonstrates the study by Pradeep et al, which is not the case. It would be better to introduce Figure 1 of the current manuscript in a more suitable place in order not to mislead the readers.

5.     Line 226: “Ayub et al. have shown that ovarian cancer stem cells ….”  - lacks a citation.

Neither reference 64 no 65 of the bibliography has Ayub as the first author.

6.     It is better to re-introduce same citations at the end of each sentence where a certain study

is mentioned. Authors often describe concrete studies in multiple sentences and do not cite them until the end of the paragraph. To a reader, it is not clear whether the statements are authors’ own thoughts vs the findings belong to previously or subsequently cited paragraph vs the citations are simply missing.

According to the reviewer’s constructive comments, we rewrote appropriately and added some references, including the report by Ayab et al. (Oncotarget. 2015;6(18):16437-48.) in the revised manuscript and in the reference list (Reference 65). 

Reviewer 2 Report

This manuscript is well organized about cancer stem cells in ovarian cancer. The authors focused on the biological mechanism which regulate cancer metastases and chemotherapy resistance. Therefore, further studies will lead to the development of innovative treatment for ovarian cancer.

Author Response

 Reviewer #2 (Reviewer Comments to the Author):

Thank you very much for the careful reading of our manuscript. We thank the reviewers for constructive suggestions that helped us improve the manuscript.

Reviewer 3 Report

This is a well-written short review article compiling existing information about the potential role of cancer stem cells driving progression of metastasis and chemoresistance in ovarian cancer.

The following are recommendations by this reviewer to improve the message of the article:

Figures 1 and 2. The legend of the figures should indicate why the molecules have been highlighted with different colors. Is there a reason for highlighting some molecules with purple and others with pink?

Line 75; delete ‘have’ in Zhang et al. have identified…

Line 83; cancer cell clones…

Line 86: …than the CD24-negative…

Line 88: delete ‘has’ in group has previously….

Line 98: used to isolate…

Line 108: delete ‘have’ in Previous studies have demonstrated…

Line 115: delete ‘have’ in Roy et al. have demonstrated…

Line 119: with the microenvironment…

Line 120: In this regard, a previous study…

Line  121: the possibility that the peritoneal…

Line 123: In addition, a recent study by…. shows (delete have shown).

Line 128: group demonstrated (delete has)

Line 128: CD44 variant 6 (space needed after CD44)

Line 130: highly enriched in CD44v6…

Line 157: our recent study demonstrated (delete has)

Line 168: implying that CD44v6…

Line 171: Bai et al showed (not have shown)

Line 187: A previous study revealed (delete has)

Line 190: Gao et al demonstrated (delete has)

Line 196: Steg et. al reported (delete have)

Line 201: delete ‘should be’ at the beginning of the line.

Line 206: we also demonstrated (delete have)

Line 226: Ayub et al. showed (not have shown)

Line 228: ALDH-1 expression is associated with unfavorable survival

Line 278: complex processes…

Line 283: A previous study reported (delete has)

Line 290: Shannon et al. demonstrated (delete have)

Line 311: Chen et al. demonstrated (delete have)

Line 314: Song et al. demonstrated (delete have)

Line 322: these findings indicate (not indicated)

Author Response

Reviewer #3 (Reviewer Comments to the Author):

1.     Figures 1 and 2. The legend of the figures should indicate why the molecules have been

highlighted with different colors. Is there a reason for highlighting some molecules with purple and others with pink?

According to the reviewer’s constructive suggestions, we changed the colors simply in Figure 1, and Figure 2. We only highlighted the cancer stem cell markers, including CD44, CD117, ALDH, CD133, CD24, and EpCAM, with red, and embryonic developmental signaling, including, Wnt, Notch, and Hedgehog, with green, in order to avoid confusion.

2.     Line 75; delete ‘have’ in Zhang et al. have identified…  

Line 83; cancer cell clones…

Line 86: …than the CD24-negative…  

Line 88: delete ‘has’ in group has previously….

Line 98: used to isolate…  

Line 108: delete ‘have’ in Previous studies have demonstrated…

Line 115: delete ‘have’ in Roy et al. have demonstrated…  

Line 119: with the microenvironment… 

Line 120: In this regard, a previous study…  

Line  121: the possibility that the peritoneal…  

Line 123: In addition, a recent study by…. shows (delete have shown).  

Line 128: group demonstrated (delete has)

Line 128: CD44 variant 6 (space needed after CD44)

Line 130: highly enriched in CD44v6…

Line 157: our recent study demonstrated (delete has)

Line 168: implying that CD44v6…

Line 171: Bai et al showed (not have shown)

Line 187: A previous study revealed (delete has)

Line 190: Gao et al demonstrated (delete has)

Line 196: Steg et. al reported (delete have)

Line 201: delete ‘should be’ at the beginning of the line.

Line 206: we also demonstrated (delete have)

Line 226: Ayub et al. showed (not have shown)

Line 228: ALDH-1 expression is associated with unfavorable survival

Line 278: complex processes…

Line 283: A previous study reported (delete has)

Line 290: Shannon et al. demonstrated (delete have)

Line 311: Chen et al. demonstrated (delete have)

Line 314: Song et al. demonstrated (delete have)

Line 322: these findings indicate (not indicated)Minor Point:

In response to these excellent suggestions, we rewrote our manuscript appropriately. Thank you very much for the careful reading of our manuscript.
